# Effects of Exercise Training on Patient-Specific Outcomes in Pancreatic Cancer Patients: A Scoping Review

**DOI:** 10.3390/cancers15245899

**Published:** 2023-12-18

**Authors:** Kim Rosebrock, Marianne Sinn, Faik G. Uzunoglu, Carsten Bokemeyer, Wiebke Jensen, Jannike Salchow

**Affiliations:** Department of Oncology, Hematology, BMT with Section Pneumology, Hubertus Wald Tumor Center—University Cancer Center Hamburg, University Medical Center Hamburg-Eppendorf, Martinistr. 52, 20246 Hamburg, Germanyma.sinn@uke.de (M.S.); f.uzunoglu@uke.de (F.G.U.); cbokemeyer@uke.de (C.B.); w.jensen@uke.de (W.J.)

**Keywords:** pancreatic cancer, exercise therapy, resistance training, aerobic training, quality of life, physical function, muscle strength, fatigue, cachexia

## Abstract

**Simple Summary:**

Pancreatic cancer remains one of the most malignant solid tumours. Due to the disease’s rapid progression, pancreatic cancer patients often experience great distress and suffer from symptoms such as pain, cachexia, sarcopenia, and cancer-related fatigue. The importance of exercise interventions as a supportive strategy in cancer patients to address these and other symptoms is steadily increasing, and strong evidence of various beneficial effects is shown. However, the impact of exercise on pancreatic cancer patients is still poorly characterized. Therefore, this scoping review aimed to evaluate the impact of exercise on pancreatic cancer patients. The results of this review suggest that exercise can improve different patient-specific outcomes in this population. To confirm these findings, further randomized-controlled trials are needed.

**Abstract:**

Background: International guidelines have already highlighted the beneficial effects of exercise in common cancer entities. However, specific recommendations for pancreatic cancer are still missing. This scoping review aimed to evaluate the impact of exercise training on patient-specific outcomes in pancreatic cancer patients. Methods: A literature search was undertaken using PubMed, Web of Science, and Cochrane Library. We included randomized controlled trials (RCTs) published before August 2023 with structured exercise interventions during or after pancreatic cancer treatment. Results: Seven articles that prescribed home-based or supervised exercise with aerobic or resistance training or both were reviewed. The results indicate that exercise is feasible and safe in pancreatic cancer patients. Furthermore, exercise was associated with improved quality of life, cancer-related fatigue, and muscle strength. Concerning other outcomes, heterogeneous results were reported. We identified a lack of evidence, particularly for patients with advanced pancreatic cancer. Conclusion: Exercise interventions in pancreatic cancer patients are feasible and can lead to improved quality of life, cancer-related fatigue, and muscle strength. However, further studies with larger sample sizes are needed to clarify the potential of exercise in pancreatic cancer, in particular for advanced stages.

## 1. Introduction

Exercise therapy is getting more attention in the supportive care of cancer patients. International guidelines and recommendations highlight the importance of aerobic and resistance exercise during and after cancer treatment [1,2,3,4]. Improvements in cardiorespiratory fitness, strength, physical function, and health-related quality of life, including anxiety and depression, have been reported [1,2,5]. However, a lot of evidence is based on intervention studies of breast, prostate, lung, and colorectal cancer patients [2].

Pancreatic cancer remains one of the most malignant solid tumours with a 5-year survival of 5–12% in all stages [6,7]. At diagnosis, only approximately 20% of patients have a resectable tumour, and the majority are diagnosed with locally advanced or metastatic pancreatic cancer. Surgical resection in combination with chemotherapy is the only potentially curative therapy [6,8]. Given the poor prognosis and low survival rates, symptom control and improvements in quality of life play a major role in supportive care [9]. Pancreatic cancer patients often suffer from numerous symptoms such as pain, cachexia, fatigue or anxiety and depression [10,11]. In particular, sarcopenia and cachexia negatively impact the situation and prognosis of pancreatic cancer patients [12,13]. To address these symptoms, exercise therapy may be used as a supportive strategy. Better symptom control and improvements in cancer-related fatigue, pain, dyspnea and insomnia have been reported in several exercise intervention trials in cancer patients [14,15]. Furthermore, recent reviews have indicated safety, feasibility, and preliminary beneficial effects in advanced cancer patients [16,17]. However, so far, there are no specific recommendations and exercise guidelines for patients with pancreatic cancer.

We therefore want to systematically summarise the effects of exercise training in pancreatic cancer patients based on randomised controlled trials (RCT). To our knowledge, two systematic reviews [18,19] and two scoping reviews [20,21] have evaluated exercise training in this cancer type. They analysed RCTs, uncontrolled intervention studies, and case reports or case series. We want to focus our review on the evidence of RCTs because they provide the highest level of evidence among clinical trials. With new upcoming randomized exercise trials in pancreatic cancer patients, we decided to only include trials with a control group, because the comparison between the intervention and the control group could be used to measure the effectiveness of the exercise intervention. Our goal is to assess the effects of exercise on different patient-specific outcomes such as health-related quality of life, cancer-related fatigue, cachexia, physical function, muscle strength, and body composition. Furthermore, we want to identify research gaps in the existing literature to develop implications for future studies.

## 2. Materials and Methods

### 2.1. Protocol

This scoping review was accomplished following the PRISMA-ScR guidelines (Preferred Reporting Items for Systematic reviews and Meta-Analyses extension for Scoping Reviews) [22], and our methodical framework was based on the five broad stages of Arksey and O’Malley [23]. Our review was not registered in PROSPERO.

### 2.2. Eligibility Criteria

To examine the studies for eligibility and define our inclusion and exclusion criteria, we used the PICOS approach (Population, Intervention, Comparator, Outcomes and Study design):Population: The included participants were adult men or women (age ≥18 years) with pancreatic cancer of any stage (I–IV). More than 80% of the study population had been diagnosed with pancreatic ductal adenocarcinoma (PDCA).Intervention: Studies were eligible if they evaluated structured exercise interventions (e.g., aerobic or resistance exercise) with or without supervision during or after pancreatic cancer treatment.Comparator: We only included studies with a control group who received usual or enhanced usual care.Outcome: As a primary outcome we evaluated the effects of exercise training on patient-specific outcomes in pancreatic cancer patients, such as health-related quality of life, cancer-related fatigue, physical function, muscle strength, body composition and cachexia.Study Design: This review only included intervention-based randomised controlled trials (RCTs).

### 2.3. Information Sources

Sources for the conducted review were PubMed, Web of Science, and Cochrane Library. Searches were carried out until 5 August 2023

### 2.4. Search

To identify the potentially relevant articles, we used the aforementioned electronic databases. The main terms for the search were as follows:

((pancreatic neoplasms[MeSH Terms]) OR (pancreatic cancer) OR (pancreatic adenocarcinoma) OR (pancreatic ductal adenocarcinoma)) AND ((exercise therapy[MeSH Terms]) OR (exercise training) OR (Exercise) OR (aerobic exercise[MeSH Terms]) OR (aerobic training) OR (resistance training[MeSH Terms]) OR (physical activity)).

Search terms were adapted to the respective databases. There were no limitations imposed during electronic searching. Results from database searches were collected using Endnote (X7.8, Clarivate, London, UK).

### 2.5. Selection of Sources of Evidence

After electronic searching and data collection, duplicates were removed automatically (using Endnote) or manually. Firstly, article titles and abstracts were screened by K.R. to remove irrelevant articles. The full text of the remaining articles was retrieved. K.R., W.J. and J.S. screened the full texts for eligibility according to the predefined inclusion and exclusion criteria. Discrepancies were solved by consensus. The selection process was recorded in detail to complete the PRISMA flow diagram (see Figure 1).

### 2.6. Data Charting Process and Data Items

Data from the included studies were charted into a previously developed Excel sheet. Extracted information included first author, study type, study year, study location, sample size, patient characteristics (age, sex, BMI), cancer stage, intervention, duration, frequency, supervision or home-based program, and any reported outcomes of interest. The charted data were discussed and continuously updated in an iterative process.

### 2.7. Synthesis of Results

After data charging, we collected, summarized, and reported the results of the included studies. During the extraction process, the different results were grouped by the reported outcome category.

## 3. Results

### 3.1. Selection and Characteristics of Sources of Evidence

A total of 1935 records were found (PubMed 714; Web of Science 877; Cochrane 344). Some 383 duplicates were removed, and the remaining 1552 articles were screened by titles and abstracts. After removal of irrelevant records, 64 full-text articles were assessed for eligibility, and a further 57 articles were excluded. Figure 1 provides detailed information on the reasons for exclusion and the selection process. In total, seven articles were included in this review [24,25,26,27,28,29,30], which included a total of 396 patients. Based on our inclusion criteria, all trials were randomized controlled trials (RCT).

The included seven articles were based on five different trials, because one trial had its results reported in three different articles [24,25,26]. The findings were published between 2012 and 2023. The number of participants ranged from 28 [26] to 151 [29], with a mean age between 51.9 [28] and 66.5 years [27]. The majority of patients were diagnosed with pancreatic ductal adenocarcinoma. The trials included pancreatic cancer patients with stage I–IV. Stage I and II were the most common stages, and only six patients (barely 2%) had metastatic/stage IV diseases. Therefore, nearly all included patients underwent surgery [24,25,26,27,28,29,30]. One trial included a preoperative exercise intervention concurrent with neoadjuvant treatment [29], and four trials included a postoperative intervention [24,25,26,27,28,30]. Most patients received adjuvant or neoadjuvant chemotherapy [24,25,26,27,28,29,30].

Mean body mass index (BMI) differed between 21.1 [28] and 28.2 [29]. The trial of Kamel et al. focused on pancreatic cancer patients with cancer-induced cachexia. They only included patients with weight loss >5% over the past six months or weight loss >2% in patients with BMI less than 20 kg/m^2^ [28]. Furthermore, about half of the study population (55.8%) of Wiskemann et al. was cachectic (with a weight loss of 10% or more in the last 6 months) [25]. In the study of Ngo-Huang et al. about 50% of the patients were sarcopenic at baseline [29]. Table 1 summarizes the study and intervention characteristics.

### 3.2. Intervention Characteristics

All trials prescribed a structured exercise intervention program. Some 396 patients were randomized to exercise or control, with a total of 207 patients in the intervention groups. The primary endpoints were feasibility [25], quality of life [27,30], cancer-related fatigue [27], different aspects of physical function and mobility [24,27,28,29], muscle strength [28] and change in body composition [26,28]. Combined resistance and aerobic training was offered in two trials [29,30]. Another two trials focused on resistance training [24,25,26,28], while the trial by Yeo et al. included aerobic walking exercise [27]. The duration of intervention ranged from 12 weeks [27,28] to 12 months [30]. Exercise sessions per week varied from two times per week [24,25,26,28] to seven times per week (at the beginning of the intense physiotherapy program of Weyhe et al.) [30]. Of the different interventions, *n* = 2 [28,30] were supervised, *n* = 2 [27,29] were home-based exercise training, and one trial prescribed both supervised (group RT1) and home-based (group RT2) exercise training [24,25,26]. Studies with home-based interventions used phone or video calls as well as diaries and questionnaires to monitor their patients [24,25,26,27,29]. In addition, Ngo-Huang et al. provided their participants with activity trackers (Fitbit Charge 2) [29]. Patients in the intervention group of Weyhe et al. received a pedometer after their supervised exercise program to encourage and continue with home-based exercise training during the 12 month follow-up [30].

In two trials, the control group received enhanced usual care [29,30]. The control group of Weyhe et al. received standard physiotherapy after surgery, whereas the intervention group received an intensified physiotherapy program and detailed exercise prescription after discharge from rehabilitation [30]. The control group of Ngo-Huang et al. received an information packet about exercise, stretching, and flexibility as well as the activity trackers, but no specific exercise prescription [29].

### 3.3. Safety and Feasibility

To evaluate the safety and feasibility of structured exercise interventions, different outcome measures should be considered. The included articles reported no exercise-related adverse events [24,25,26,30] or did not provided explicit information [27,28,29]. Most studies reported a drop-out rate of around 30% or less [24,25,27,28,29]. Only Weyhe et al. reported a drop-out rate of 36% for their intervention group and 43% for their control group [30]. The authors described different reasons for drop-out, such as disease progression or death, loss to follow-up, treatment elsewhere, withdrawal, and discontinued intervention [24,25,26,27,28,29,30].

In total, four articles assessed exercise adherence [24,25,26,30]. Reported overall adherence varied between 59.2% [25] and 80% [30]. But in the study of Steindorf et al., the training adherence dropped steadily during the intervention period [24]. Although Ngo-Huang et al. provided no direct adherence data, they reported maintenance of average weekly activity time in their study [29].

### 3.4. Quality of Life

Four studies investigated the influence of exercise on quality of life using different scales [24,27,29,30]. Two studies used the European Organization for Research and Treatment of Cancer C30 questionnaire (EORTC QLC-C30) and the pancreatic cancer-specific module (EORTC PAN26) [24,30]. One study used the Short Form Health Survey SF-36v2 [27] and another study the SF-8 [30]. Ngo-Huang et al. used the Functional Assessment of Cancer Therapy—Hepatobiliary questionnaire (FACT-hep) [29].

Three of the four studies investigating quality of life reported significant improvements in different domains or subscales. Steindorf et al. reported significant between-group differences in favour for resistance training for global quality of life, cognitive functioning, physical functioning, and sleep problems after three months, but not after six months [24]. Weyhe et al. found significant differences between cohorts for role functioning (from 6–12 months postoperatively) and physical functioning (from 3–12 months postoperatively), as measured with EORTC QLC-C30 [30]. Improvement in overall quality of life and other subscales was not reported [30]. Yeo at al. observed significantly improvements in quality of life in both the intervention and control group [27]. For the intervention group, the post-intervention scores were significantly improved in six domains; both the physical and mental component scores improved [27]. For the control group, only four domains and the mental component score significantly improved [27]. In contrast, Ngo-Huang et al. reported no statistically significant changes or between-group differences in health-related quality of life [29]. The main results of the included studies are presented in Table 2.

### 3.5. Cancer-Related Fatigue

Three studies assessed cancer-related fatigue [24,27,30]. Steindorf et al. used the Multidimensional Fatigue Inventory (MFI) [24]. Yeo et al. investigated fatigue using the Fatigue Visual Analog Scale (FVAS) and the Functional Assessment of Chronic Illness Therapy-Fatigue (FACIT-F) scale [27]. Weyhe et al. used the fatigue subscale of the EORTC QLC-C30 questionnaire [30].

Two of the three studies investigating fatigue reported significant improvements. Patients in the intervention group of Yeo et al. had significant improved fatigue scores (FACIT-F scale and FVAS) at study completion [27], whereas the control group had no significant improvements compared to baseline scores [27]. Steindorf et al. reported significant improved physical fatigue for the intervention group compared to the control group after three months, but not after six months [24]. In the study by Weyhe et al., no significant between-group differences in the fatigue subscale were reported [30]. However, for all post-hospitalization visits, the intervention group showed a clinically relevant (i.e., >10 points) lower symptomatic burden compared to comparison group [30].

### 3.6. Physical Function

Five studies assessed physical function by using various outcome measures [25,27,28,29,30]. Cardiorespiratory fitness was measured by laboratory-based cardiopulmonary exercise testing (CPET) [25] or performance-based tests like 400 m walk test (400m-WT) [28] or 6 min walk distance (6MWD) [29]. Two studies assessed functional mobility and gait speed via a 6 m usual and fast walk test (6m-WT) [28] or a 3 m walk test (3m-WT) [29]. Furthermore, two studies used 5-times sit-to-stand test (5xSTS) [28,29]. Weyhe et al. used the combined Short Physical Performance Battery (SPPB) [30], which includes evaluation of balance, gait, strength and endurance [31]. Yeo et al. assessed the length of time and approximate distance during and at the end of their walking program [27].

Three of the five studies reported significant improvements in different physical function parameters [27,28,29]. Kamel et al. reported significant improvements in the 400 m walk test (400m-WT), 6 m usual walk test (6m-WT) and the 5-times sit-to-stand test (5xSTS) in the intervention group compared with the control group [28]. However, no significant difference in the 6 m fast walk test was observed [28]. In the study by Ngo-Huang et al., participants of both groups showed a statistically and clinically significant increase in 6 min walk distance (6MWD) [29]. In addition, the participants of the intervention group had a statistically significant improvement in 5xSTS time and 3 m walk test (3m-WT) [29]. However, none of these changes were significantly different between both groups [29]. In the study by Yeo et al., participants in the intervention group were walking twice as far, and were significantly more likely to have continued walking or another form of exercise at study completion compared with the control group [27]. In contrast, Weyhe et al. reported no significant influence of the exercise intervention on physical performance measured with the SPPB [30]. With regard to cardiopulmonary exercise testing (CPET), most parameters showed no between-group differences in the study by Wiskemann et al. [25]. Only the supervised resistance training group showed a significant improvement in peak work rate compared with both the control group and the home-based resistance training group [25].

### 3.7. Muscle Strength

Three studies assessed muscle strength [25,28,29]. Therefore, they used isokinetic and isometric dynamometry [25,28] and/or hand-held dynamometry [25,29]. Furthermore, Ngo-Huang et al. measured the number of bicep curls performed in a 30 s period with the arm curl test [29].

All studies reported significant improvements in at least one muscle strength parameter. In the study by Kamel et al., the intervention group showed a significant increase in maximum isokinetic peak torque of knee extensors, elbow flexors and elbow extensors following the 12-week resistance training compared to the control group [28]. Similarly, they reported a significant improvement in the maximum voluntary isometric contraction of the knee and elbow flexors and extensors in favour of the intervention group [28]. Wiskemann et al. also reported improved muscle strength [25]. Their observed strength gain was always higher in the supervised resistance training group than the home-based group [25]. They reported increased maximal isokinetic peak torque for elbow flexors and extensors for the supervised intervention group compared to the control group, as well as compared to the home-based intervention group [25]. For the maximum voluntary isometric contraction, there were statistically significant differences in elbow flexors and knee extensors for the supervised intervention group compared with the control group, and in knee extensors for home-based intervention group compared with the control group [25]. Using hand-held dynamometry, the authors reported significant differences in knee extensors for the home-based intervention group compared with the control group and in knee flexors when comparing the two intervention groups [25]. In the study by Ngo-Huang et al., both groups showed a statistically significant increase in arm curl test repetitions [29]. However, no improvement in handgrip strength was observed [29].

### 3.8. Body Composition and Muscle Mass

Three studies assessed different body composition parameters [26,28,29]. Dual-energy X-ray absorptiometry (DEXA) was used to evaluate fat mass percent and lean body mass in one study [28]. Two studies used computer tomography (CT scans) to assess body composition [26,29]. Only one of the three studies reported significant effects of exercise training on body composition. Kamel et al. observed a significant increase in the lean mass of the upper limb, lower limb, and appendicular skeletal muscles in the intervention group compared to the control group [28]. In contrast, Ngo-Huang et al. showed no significant changes in body composition measured by skeletal muscle index (SMI) and density (SMD) in their study population [29]. Similarly, Wochner et al. observed no significant changes in muscle and adipose tissue compartments [26]. However, they found a moderate-to-high correlation between muscle–area and muscle strength parameters [26]. Furthermore, the authors also investigated prognostic parameters for overall survival [26]. Both a high visceral-to-subcutaneous-fat ratio and loss of muscle mass were predictors of poor overall survival in their study [26].

## 4. Discussion

This scoping review summarizes the evidence of exercise interventions in pancreatic cancer patients. We focused our review on the evidence of RCTs to assess the effects of exercise on different patient-specific outcomes such as health-related quality of life, cancer-related fatigue, cachexia, physical function, muscle strength, and body composition.

Five prospective randomized trials with in a total of 396 patients fulfilled the predefined criteria for further analysis. All but six patients (corresponding to 1.5% of patients) had localized diseases. In all five trials, surgery was part of the cancer-specific treatment for most of the patients, and exercise was undertaken preoperatively/as a neoadjuvant in one trial and postoperatively in four trials. The results of the included randomized controlled trials show that exercise training is feasible and safe, with no adverse events reported. In terms of patient-specific outcomes, beneficial effects on quality of life, fatigue and muscle strength were found.

### 4.1. Quality of Life

International guidelines about exercise in cancer patients have already highlighted the potential of physical activity in supportive cancer care to reduce symptoms and improve quality of life (QoL) [1,2,3]. However, they have not yet published specific recommendations for pancreatic cancer patients. In our scoping review, the available data show that exercise can improve several dimensions of QoL, e.g., cognitive functioning, role functioning, sleep problems, social functioning or mental health [24,27,30]. Only one of the four trials investigating QoL reported no significant changes [29]. A possible explanation for the differences in the effectiveness of the interventions could be the inconsistencies in the measurement scales and QoL dimensions. Furthermore, Steindorf et al. reported significant improvements in global quality of life, cognitive functioning, and sleep problems after three months, but not after six months [24]. The authors suggest that decreasing adherence could be a possible reason [24]. Due to their findings, Steindorf et al., Yeo et al., and Weyhe et al. recommended exercise therapy for pancreatic cancer patients [24,27,30].

### 4.2. Cancer-Related Fatigue

Cancer-related fatigue is another important target of supportive cancer care and directly related to health-related quality of life [9]. Most pancreatic cancer patients experience fatigue, with prevalence ranging from 55% to 94% [32,33,34] and those reporting fatigue are more likely to report functional impairments [34]. Our findings indicate that exercise may reduce cancer-related fatigue in pancreatic cancer patients. Two studies reported significant improvements in fatigue [24,27], while Weyhe et al. showed no significant but clinically relevant differences for all post-hospitalization visits [30]. The differences in the effectiveness of fatigue reduction between the three studies could be explained by the heterogeneity of intervention type and duration, as well as the larger sample size of Yeo et al. Nevertheless, our findings are consistent with those of a Cochrane review, which showed the beneficial effects of exercise interventions on fatigue, especially for common cancer types like breast and prostate cancer [14]. However, they only reported significant effects for aerobic exercise [14]. In our review, the type of exercise (resistance and/or aerobic training) varied between the trials examining fatigue. Therefore, further research is required to determine the optimal exercise type.

### 4.3. Physical Function

Regarding the effect of exercise on physical function, mixed effects were observed. Three out of five studies showed statistically or clinically significant improvements in various scales (6MWD, 400m-WT, 6m-WT, 3m-WT, 5xSTS, and mean average walk distance) [27,28,29]. Ngo-Huang et al. reported improved 6MWD in both the intervention and control groups [29]. According to the authors, the extrinsic motivation to exercise from the health care team, but also the intrinsic motivation by participating in the study may be one possible explanation. With this background, it may become more difficult for future studies to show significant between-group differences. Another explanation for the heterogeneous results could be the different time of assessment and the variability of the measurement scales. While some of the studies performed physical tests, others used questionnaires to assess physical function. To confirm the positive effects on physical function and improve comparability, greater consistency in outcome measurement may be helpful. For cardiorespiratory fitness parameters, Wiskemann et al. reported an increase in peak work rate in the supervised resistance training group, but no improvement in other parameters [25], so the potential of exercise to improve cardiorespiratory fitness remains unclear. Further studies using cardiopulmonary exercise testing in both resistance and aerobic training are needed.

### 4.4. Muscle Strength

In terms of muscle strength, results suggest that exercise may be effective in improving muscle strength in pancreatic cancer patients. Two studies reported gains in the muscle strength of different muscle groups as measured by isokinetic and isometric dynamometry [25,28]. Furthermore, Ngo-Huang observed increased arm curl test repetitions [29].

### 4.5. Cachexia, Sarcopenia, and Body Composition

Improving muscle strength is of particular interest in view of the common and significant concerns of cancer-related cachexia and sarcopenia in pancreatic cancer patients. Cachexia is present in up to 80% of pancreatic cancer patients, and both cachexia and sarcopenia are associated with poor prognosis [12,13,35]. The study of Kamel et al. showed improved muscle strength and increased lean mass of the upper limb, lower limb and appendicular skeletal muscles in cachectic patients [28]. Although two other studies also reported about half of their study population being cachectic [25] or sarcopenic [29], they did not present separate analyses or results for this subgroup.

Furthermore, the current evidence on body composition and muscle mass is heterogeneous, with only one [28] out of three studies showing improvement [26,28,29]. All of the three studies prescribed resistance training interventions, but the intervention duration and date of assessment differed between the studies. Only Kamel et al. showed a significant increase in lean mass. A possible explanation could be that they assessed body composition at three months, whereas the other trials chose a later assessment date. Therefore, further research is needed to specify the effects of exercise on these outcomes and to make recommendations for pancreatic cancer patients with cachexia. Similarly, a recent Cochrane review did not find enough evidence for exercise in cancer cachexia, and therefore highlighted the need for future studies [36].

### 4.6. Feasibility

Our scoping review reveals that exercise is generally feasible in the frail population of pancreatic cancer patients. Adherence was reported as high, with rates between 60–80% in studies that directly measured this outcome [24,25,26,30]. Drop-out rates were reported at around 30% or less [24,25,27,28,29]. However, adherence dropped steadily in the study of Wiskemann et al. [24,25,26], which might be explained in part by the overall poor prognosis of pancreatic cancer patients and the frequently rapid progression of the disease.

This may also be one of the reasons why the trials that could be included in this review mainly focused on patients with localised pancreatic cancer. However, ongoing trials are already working to fill this research gap. At the annual congress of the American Society of Clinical Oncology (ASCO) in 2022, Neuzillet et al. presented the first favourable results of their APACaP-trial (adapted physical activity in patients with advanced pancreatic cancer) [37]. They reported improvements in several dimensions of health-related quality of life [37]. However, final results have not been published yet.

### 4.7. Limitations and Strengths

Our review has different limitations. First, we could only include seven articles (based on five trials) comprising 396 patients overall. Most of the trials had a small sample size. As already mentioned, the majority of the study population was diagnosed with stage I or II pancreatic cancer. Therefore, evidence for locally advanced and metastatic pancreatic cancer remains sparse. Secondly, we found a considerable heterogeneity in exercise prescription and in the measurement of outcomes. The observation period of the included studies was heterogeneous and insufficient to assess long-term effects. The endpoints analysed and the methods used were rather inhomogeneous, limiting the comparability of the data. These are possible explanations for the differences in effectiveness between the included studies. To improve comparability, future studies need standardized exercise prescriptions, for example, based on the Exercise Guidelines for Cancer Survivors [1]. Furthermore, following the scoping review approach, we did not perform a full systematic review or a systematic critical appraisal of the included sources of evidence.

The strengths of this review must also be acknowledged. We used the standardized PRISMA-ScR methodology to map the evidence on exercise in pancreatic cancer and to identify research gaps. Moreover, we included only RCTs, which provide the most reliable evidence compared to single-arm trials. At the same time, most of our findings are consistent with those of previous reviews in pancreatic cancer patients, including RCTs, single-arm trials, case reports and case series [19,20].

This consistency supports the potential beneficial effects of exercise in pancreatic cancer. However, there is a need for further RCTs with larger sample sizes to prove this potential. For future studies, we recommend considering or examining the aspects presented in Figure 2.

## 5. Conclusions

This scoping review examines the effects of exercise interventions in pancreatic cancer patients. Exercise is feasible and safe in this patient group. Existing evidence suggests that exercise can improve quality of life, cancer-related fatigue, and muscle strength. However, only a few RCTs with small sample sizes were available for pancreatic cancer patients. There is a lack of evidence, particularly for patients with advanced pancreatic cancer.

Furthermore, the effects of exercise in patients with cancer-related cachexia or sarcopenia still need to be better understood. Therefore, further RCTs with larger sample sizes are needed to investigate the potential of exercise in pancreatic cancer. To improve comparability and validity of the different outcomes, it would be helpful to have greater consistency in the implementation and duration of the intervention, as well as the measurement of outcomes.

## Figures and Tables

**Figure 1 cancers-15-05899-f001:**
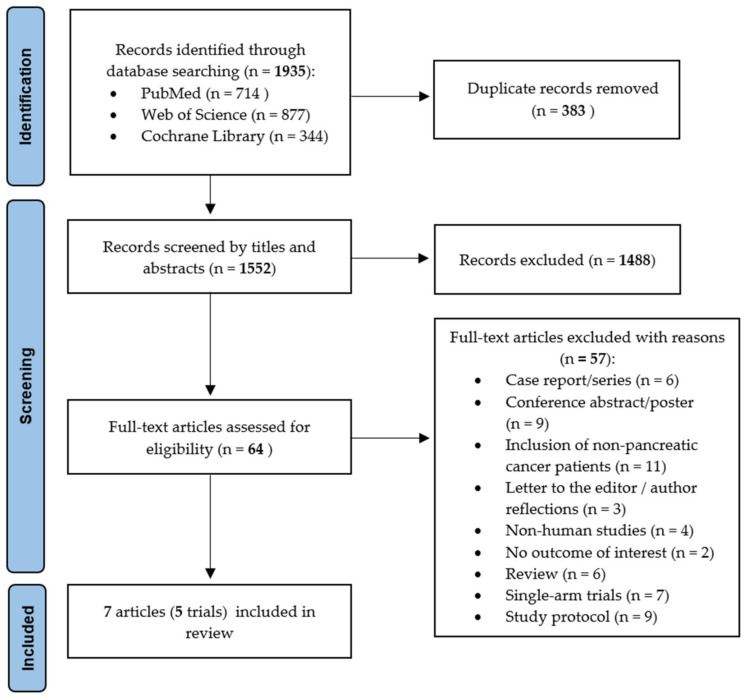
Preferred Reporting Items for Systematic Reviews and Meta-Analysis (PRISMA) flow diagram with a number of studies identified and selected for inclusion in the scoping review [22].

**Figure 2 cancers-15-05899-f002:**
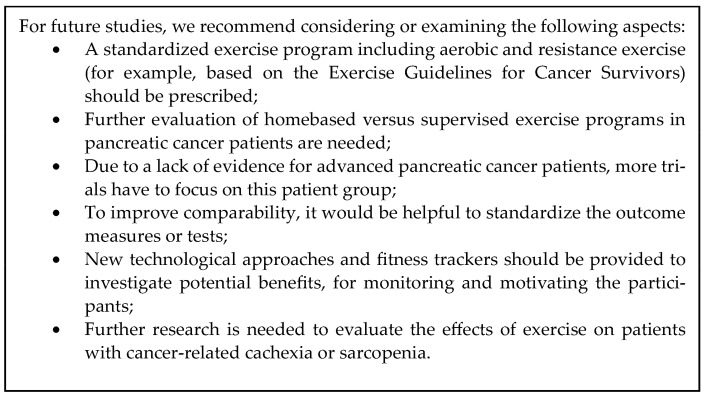
Implications for future research [1].

**Table 1 cancers-15-05899-t001:** Study and intervention characteristics.

Author (Year), Study Location	Sample Size (IG, CON)	Cancer Type	Cancer Stage	Patient Characteristics (Age, % Male, BMI)	Intervention	Supervision	Cancer Treatment
Kamel et al. (2020), Egypt [28]	40 (20 IG, 20 CON)	87.5% PDCA, 7.5% dCCA, 5% ampullary	Stage I–IV (97.5% I–II) (stage IV *n* = 1)	51.9 years, 65% male, BMI 21.1	RT (2x/wk., duration: 12 wks., beginning: 3 mth post-surgery)	supervised	77.5% OP + ACT, 17.5% NCT + OP, 5% NCT + OP + ACT
Steindorf et al. (2019), Germany (SUPPORT-Study) [24]	47 (9 IG1, 21 IG2, 17 CON)	87.2% PDCA, 10.6% dCCA, 2.1% ampullary ductal	Stage I–IV (mostly stage II) (stage IV *n* = 1)	60.5 years, 53.2% male, BMI 23.7	RT (60 min 2x/wk., duration: 6 mth post-surgery or no surgery)	supervised (IG1) or home-based (IG2)	89.4% OP + ACT, 4.3% NCT + OP + ACT, 4.3% NCT + OP + ACT, 2% only CT
Wiskemann et al. (2019), Germany (SUPPORT-Study) [25]	43 (9 IG1, 20 IG2, 14 CON)	88.4% PDCA, 9.3% dCCA, 2.3% ampullary ductal	(see above)	60.4 years, 55.8% male, BMI 23.3	(see above)	(see above)	88.4% OP + ACT, 4.7% NCT + OP + ACT, 4.7% NCT + OP + ACT, 2.2% only CT
Wochner et al. (2020), Germany (SUPPORT-Study) [26]	28 (19 IG, 9 CON)	85.7% PDCA, 10.7% dCCA, 3.6% ampullary ductal	(see above)	62.1 years, 62.3% male, BMI 23.9	(see above)	(see above)	85.7% OP + ACT, 7.1% NCT + OP + ACT, 3.6% NCT + OP + ACT, 3.6% only CT
Weyhe et al. (2022), Germany [30]	56 (28 IG, 28 CON)	85.7% AdenoCA, 7.1% NET, 5.4% IPMN, 1.8% acinar cell carcinoma	Stage I–IV (mostly stage II) (stage IV *n* = 4)	66.4 years, 58.9% male, BMI 26.8	RT + AT (post-surgery intensified rehabilitation) (beginning 24 h post-surgery: 3x/d in-bed cycling, second week 3x/d 15 min walking + muscle exercises 5x/wk., after discharge 3x/wk. 15–20 min RT + walking program; duration: 12 mth)	supervised, after discharge from reha-bilitation clinic home-based	all received surgery, 72.5% received CT, RTx or both
Yeo et al., (2012), USA [27]	102 (54 IG, 48 CON)	91.2% PDCA, 2.9% bile duct cancer, 3.9% IPMN, 1.9% duodenal cancer	Stage I–III (mostly stage II)	66.5 years, 50% male, BMI 27	postresection walking program (AT) (3–5x/wk. 20–40 min, duration: 12 wks.)	home-based	all received surgery, 69.6% received CT
Ngo-Huang et al. (2023), USA [29]	151 (75 IG, 76 enhanced usual care)	Pancreatic cancer	57% potentially resectable, 33% borderline resectable, 10% locally advanced	66.2 years, 60.9% male, BMI 28.15	RT + AT preoperative (AT: ≥30 min ≥ 5x/wk. moderate-intensity, RT: ≥2x/wk., duration: 22 wks. CON, 24 wks. IG)	home-based	neoadjuvant therapy: 53% CT only, 4% CT + RTx, 43% both

Abbreviations: adjuvant chemotherapy (ACT), aerobic training (AT), body mass index (BMI), chemotherapy (CT), control group (CON), day (d), distal cholangiocarcinoma (dCCA), intervention group (IG), intraductal papillary mucinous neoplasm (IPMN), month (mth), neoadjuvant chemotherapy (NCT), neuroendocrine tumor (NET), operation/surgery (OP), pancreatic ductal adenocarcinoma (PDCA), radiotherapy (RTx), resistance training (RT), week (wk).

**Table 2 cancers-15-05899-t002:** Summary of study results.

Author (Year)	QoL	Physical Function	Muscle Strength	Body Composition, Muscle Mass	Measurement	Primary Endpoints
Kamel et al. (2020), [28]	not reported	Physical function ↑: significant improvement IG vs. CON: 400m-WT, usual 6m-WT, 5xSTS	Muscle strength ↑: significant improvement IG vs. CON: peak torque of knee extensors, elbow flexors/extensors	Lean mass ↑: significant improvement IG vs. CON: lean mass of the upper limb, lower limb and appendicular skeletal muscle	400m-WT, 6m-WT, 5xSTS, isokinetic and isometric dynamometer, dual-energy X-ray absorptiometry (DEXA)	Mobility, muscle strength and lean body mass after 12 weeks
Steindorf et al. (2019), (SUPPORT-Study) [24]	QoL after 3 month ↑: significant difference for month 3 (T1) but not month 6 (T2) IG vs. CON: improvement in global QoL, cognitive functioning, physical functioning, sleep problems and physical fatigue	not reported	not reported	not reported	EORTC QLQ-C30, EORTC-PAN26, MFI	Physical functioning at 6 months (subscale of EORTC)
Wiskemann et al. (2019), (SUPPORT-Study) [25]	not reported	CPET ↔ (most parameters showed no between-group differences, peak work rate ↑ IG1 vs. CON and IG1 vs. IG2)	Muscle strength ↑: IG1 vs. CON ↑ elbow flexors/extensor, knee extensor strength, IG2 vs. CON ↑ knee extensor strength, IG1 vs. IG2 ↑ elbow flexors/extensor, knee flexors	not reported	Isokinetic and handheld dynamometer, CPET, 6MWD (not reported)	Feasibility of the resistance training intervention
Wochner et al. (2020), (SUPPORT-Study) [26]	not reported	not reported	(but strong correlation between muscle strength and muscle mass)	Body composition ↔: no between-group differences at 6 months on muscle and adipose tissue parameters	CT-based measurement of adipose and muscle parameters, isokinetic dynamometer	Impact of progressive resistance training on muscle and adipose tissue compartments
Weyhe et al. (2022), [30]	QoL ↑, Fatigue ↔: EORTC QLC-C30: significant improvement IG vs. CON in physical functioning (mth 3–12) and role functioning (month 6–12), no significant differences in SF-8 and EORTC-PAN26	Physical function ↔: no significant between-group difference in physical performance (SPPB), but IG almost regain their physical condition comparable with before surgery	not reported	not reported	EORTC QLQ-C30, EORTC-PAN26, SF-8, SPPB (Short physical performance battery: Balance Test, Gait Speed Test and 5xSTS)	QoL after 12 months (measured by SF-8, EORTC QLC-C30/QLC-PAN26)
Yeo et al., (2012), [27]	QoL & Fatigue ↑: significant improvement in FACIT-FS and FVAS (IG vs. CON), SF-36 health survey: IG: significant improvement in 6 domains, physical and mental component score ↑, CON: significant improvement in 4 domains, mental component score ↑	Walk distance ↑: IG walked twice as far as CON at the end of the study, IG were significantly more likely to still be walking or engaged in another form of exercise	not reported	not reported	Fatigue and Pain Visual Analog Scale (FVAS and PVAS), FACIT-Fatigue Scale, SF-36v2	Cancer-related fatigue, physical function and QoL after 3–6 months of study participation
Ngo-Huang et al. (2023), [29]	QoL ↔: no significant improvement in QoL (FACT-Hep) or self-reported physical functioning (PROMIS) in both groups	Physical function ↑: significant improvement in 6MWD in both arms, significant improvement in 5xSTS and 3m-WT in IG, no significant between-group difference	Muscle strength ↑: both arms statistically significant improvement in arm curl repetitions, no significant improvement in handgrip strength	Body composition ↔: No statistically significant changes in SMI, SMD	FACT-hep, PROMIS 12a, 6MWD, 5STS, arm curl test, handgrip strength, 3m-WT, muscle parameter (SMI, SMD)	Change in 6MWD between enrolment and preoperative follow-up

Abbreviations: Cardiopulmonary exercise testing (CPET), control group (CON), European Organization for Research and Treatment of Cancer C30 questionnaire (EORTC QLQ-C30) and the pancreatic specific module (EORTC-PAN26), Fatigue Visual Analog Scale (FVAS), Functional Assessment of Cancer Therapy—Hepatobiliary questionnaire (FACT-hep), Functional Assessment of Chronic Illness Therapy-Fatigue Scale (FACIT-FS), Five-times sit-to-stand test (5xSTS), 400 m walk test (400m-WT), intervention group (IG), Multidimensional Fatigue Inventory (MFI), Pain Visual Analog Scale (PVAS), Patient-Reported Outcomes Measurement Information System (PROMIS), quality of life (QoL), Short Form Health Survey 8 (SF-8), Short Form Health Survey 36 (SF-36v2), Short Physical Performance Battery (SPPB), Six meter walk test (6m-WT), Six Minute Walk Distance (6MWD), skeletal muscle index (SMI), skeletal muscle density (SMD), Three meter walk test (3m-WT), improvement or significant change in favour of the intervention group therapy (↑), no significant change (**↔**).

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
