# Peer review of "Effects of Exercise Training on Patient-Specific Outcomes in Pancreatic Cancer Patients: A Scoping Review"

_cancers, 2023, doi:10.3390/cancers15245899_

Round 1

Reviewer 1 Report

Comments and Suggestions for Authors

It is assumed that it would be very difficult to make certain recommendations on this topic as a systematic review. This is due to the rapid progressive characteristic of pancreatic cancer and the fact that many patients are in a cachexia situation. In fact, all of the presented papers are based on a small sample size, and the observation period of the studies is insufficient.

It would be the best possible effort on a difficult subject. In all, this is a valuable report on a challenging topic.

I would like to suggest a revision.

It is unclear how the authors would explain the differences in the effectiveness of the rehabilitation between reports. This point should be written in more detail.

Author Response

Response to Reviewer 1:

  • As you concluded in your comment, all of the presented papers are based on small sample sizes and the observation period of the studies is insufficient. Therefore, it is difficult to make certain recommendations. In response to your feedback, we have edited our discussion and conclusion section to highlight these limitations (see Word document).
  • Futhermore we changed the title to clearly state that it is a scoping review.
  • To explain the differences in the effectiveness of the intervention between the included reports in more detail, we added explainations in the different sections of the discussion (see Word document). There are different reasons to mention such as inconsistency and heterogenity in outcome measurement and intervention characteristics (e.g. type of intervention, duration, primary outcome, etc.).

Reviewer 2 Report

Comments and Suggestions for Authors

Cancers review

Thank you very much for the opportunity to review the manuscript.

The manuscript is very well written.

I have a few suggestions to improve the submitted manuscript.

<Title>

Comment 1:

As the authors concluded, the majority of the study population was diagnosed with 397 stage I or II pancreatic cancer. The authors can mention that the results in the title.

Comment 2:

The title should clearly state that it is a scoping review.

<Introduction>

Comment 3:

The authors focused on the evidence of RCT. But why? Please clarify why the previous review study is insufficient and describe the rationale to focus on the RCT.

Author Response

Response to Reviewer 2:

  • Comment 1: As the authors concluded, the majority of the study population was diagnosed with 397 stage I or II pancreatic cancer. The authors can mention that the results in the title

Response 1: As you mentioned in your comment, the majority of the study population was diagnosed with stage I or II cancer. We considered this important finding during our editing process. However, due to the variability of the different studies, we decided not to mention this in the title. Some of the studies in our review also included higher cancer stages (e.g. Ngo-Huang et al. (n = 151) included 33% with borderline and 10% with locally advanced cancer). Therefore, we do not want to exclude these results by changing the title.

  • Comment 2: The title should clearly state that it is a scoping review.

Response 2: Thank you for your comment. We changed the title to clearly state that it is a scoping review.

  • Comment 3: The authors focused on the evidence of RCT. But why? Please clarify why the previous review study is insufficient and describe the rationale to focus on the RCT.

Response 3: We focused on the evidence of RCTs because they provide the highest level of evidence among clinical trials. Previous review articles also included single-arm trials and case reports due to a lack of RCTs in the past. With new upcoming randomized exercise trials in pancreatic cancer patients, we decided to only include trials with a control group, because the comparison between the intervention and the control group could be used to measure the effectiveness of the exercise intervention. We added this explanation in the introduction of our scoping review. 

Reviewer 3 Report

Comments and Suggestions for Authors

The authors have conducted a comprehensive systematic review examining the impact of exercise training on patient-specific outcomes in individuals diagnosed with pancreatic cancer. Their approach involved the initial identification of 1935 relevant records, leading to the inclusion of 7 articles that reported findings from 5 randomized controlled trials, all published between 2012 and 2023. The authors compiled the reported data into an Excel file and summarized the key criteria in Table 1.

            The findings of this review reveal that exercise interventions have the potential to enhance various aspects of the patient experience, including improvements in quality of life, reduction of cancer-related fatigue, and enhancement of muscle strength. However, it is important to note that the evidence supporting these positive effects is more substantial in certain contexts, and the benefit for individuals with advanced pancreatic cancer remains less clear, especially concerning cachexia or sarcopenia.

            Based on the obtained data the authors provide guidance for future randomized clinical trials in the form of a table containing concise bullet points outlining essential aspects that should be examined. This proactive approach to research recommendations enhances the practical utility of their systematic review.

 Minor suggestion for improvement would be to provide the full name of abbreviations like "MeSH Term" when first mentioned, enhancing clarity and accessibility for readers.

Author Response

Response to Reviewer 3

We appreciate the time and effort that you have dedicated to providing your valuable feedback on our manuscript. The changes in the manuscript were highlighted by using the track changes mode in MS Word. Here is a point-by-point response to your comments and concerns:

  • For reader convienience we added an abbreviation list at the end of the main text.

Reviewer 4 Report

Comments and Suggestions for Authors

The review titled "Effects of Exercise Training on Patient-Specific Outcomes in Pancreatic Cancer Patients" evaluates the impact of exercise on specific outcomes in pancreatic cancer. Seven articles were thoroughly reviewed, focusing on exercise feasibility, safety, improved quality of life, reduced fatigue, and enhanced muscle strength in pancreatic cancer patients. Additionally, the authors have outlined both the strengths and weaknesses of their study, which I believe offers a robust framework for future research. Overall, this review is well-written; the authors extensively covered the literature on exercise effects in pancreatic cancer patients and succinctly outlined the outcomes. I have a couple of suggestions that could enhance the publication's quality:

1.       Consider including an abbreviation list for reader convenience.

2.       I suggest emphasizing the impact of each selected parameter (e.g., muscle strength, physical function, etc.) in the discussion section and/or presenting a table. This approach would aid reader comprehension and provide expert insights.

Author Response

Response to Reviewer 4:

  • Comment 1: Consider including an abbreviation list for reader convenience.

Response 1: For reader convienience we added an abbreviation list at the end of the main text.

  • Comment 2: I suggest emphasizing the impact of each selected parameter (e.g., muscle strength, physical function, etc.) in the discussion section and/or presenting a table. This approach would aid reader comprehension and provide expert insights

Response 2: To emphasize the impact of each selected parameter (e.g. muscle strength, physical function, etc.), we restructured the discussion section and added subheadings. Furthermore, the mentioned parameters are summarized in table 2.
